# Interactions between Sugarcane Leaf Return and Fertilizer Reduction in Soil Bacterial Network in Southern China Red Soil

**DOI:** 10.3390/microorganisms12091788

**Published:** 2024-08-29

**Authors:** Yufeng Liu, Dan Liang, Jincheng Xing, Ziyan Xue, Zhenhua Zhang

**Affiliations:** 1Institute of Agricultural Resources and Environment, Guangxi Academy of Agricultural Sciences/Guangxi Key Laboratory of Arable Land Conservation, Nanning 530007, China; gxliuyufeng@foxmail.com; 2Guangxi Vocational College of Water Resources and Electric Power, Nanning 530023, China; 3Institute of Jiangsu Coastal Agricultural Sciences, Yancheng 224002, China; 4The School of Agriculture and Environment, The University of Western Australia, Crawley, WA 6009, Australia

**Keywords:** crop residue return, chemical fertilizer, maize, soil bacteria diversity, subtropical

## Abstract

Microbes may play an important role in the sugarcane leaf degradation and nutrient conversion process. Soil bacterial communities are more or less involved in material transformation and nutrient turnover. In order to make better use of the vast sugarcane leaf straw resources and reduce the overuse of chemical fertilizers in the subtropical red soil region of Guangxi, a pot experiment, with three sugarcane leaf return (SLR) amounts [full SLR (FS), 120 g/pot; half SLR (HS), 60 g/pot; and no SLR (NS)] and three fertilizer reduction (FR) levels [full fertilizer (FF), 4.50 g N/pot, 3.00 g P_2_O_5_/pot, and 4.50 g K_2_O/pot; half fertilizer (HF), 2.25 g N/pot, 1.50 g P_2_O_5_/pot, and 2.25 g K_2_O/pot; and no fertilizer (NF)], was conducted to assess the interactions of different SLR amounts and chemical FR levels in the soil bacterial network and the relationship between the soil properties and bacterial network by using Illumina Miseq high-throughput sequencing technology. According to the results of the soil bacterial community compositions and diversity, the soil bacterial network was changed during maize growth. SLR exerted a stronger effect on soil bacterial function than FR. Returning the sugarcane leaf to the field increased the diversity of the soil bacteria network. The bacterial communities were consistently dominated by *Acidobacteria*, *Actinobacteria*, and *Bacteroidetes* across all treatments, among which *Actinobacteria* was the most abundant bacteria type by almost 50% at the phylum level. The analysis results of the experimental factor on maize growth showed that the effect of SLR was lower than that of FR; however, this was opposite in the soil bacterial community structure and diversity. The soil bacterial network was significantly correlated with the soil total K, available N and organic matter contents, and EC. The soil bacteria community showed different responses to SLR and FR, and the FF in combination with FS partly increased the complexity of the soil bacteria network, which can further benefit crop production and soil health in the red soil region.

## 1. Introduction

Sugarcane leaf return to the field (SLR) has been recognized and implemented not only as a simple and effective straw utilization means, but also as a technique that can enhance soil organic carbon content and improve soil fertility [1,2]. Therefore, the transition from burning and discarding sugarcane residues to SLR has significantly preserved the ecological environment and improved the sustainability of the sugarcane production chain and soil health. The soil microbial network played an important role in sugarcane leaf decomposition and the turnover processes. The classical plate cultivation methods of the microorganism only reflected the cultivable soil microbe, which accounts for only 1% and less of the total soil microbial population [3]. Moreover, PCR-DGGE (Denaturing Gradient Gel Electrophoresis) and PCR-TGGE (Temperature Gradient Gel Electrophoresis) are not suitable for the soil bacterial analysis because of the error associated with PCR and their dependence on high-fidelity polymerase [4,5]. Early studies on the soil microbial network have been limited by the relatively backward technical conditions. Due to the complexity of the soil microbe and the limitation of the research method, the information on the temporal and spatial variation of the soil microbial network was limited after sugarcane leaf return in the subtropical red soil region.

Fertilization, as a common and global agricultural management strategy, constitutes the major source of nutrients in an agriculture system, which contributes the growth rate of the global grain yield by 50–60% [6,7]. However, the unreasonable consumption of chemical fertilizer has been applied to croplands to obtain a high crop yield to maintain the need for a population explosion, while the application of chemical fertilizer was often far greater than the crop demand [8]. The long-time overuse of chemical fertilizers in croplands had numerous negative environmental impacts, such as greenhouse gas excess emission [9], soil degradation [10], soil acidification [11], agricultural non-point source pollution [12], a decrease in soil biodiversity [13], and so on. Hence, optimizing fertilization management practices and mineral fertilizer reduction to minimize adverse environmental impacts while increasing crop yield is vital. Sufficient field experiments have proven that a combination of straw return and fertilizer reduction can help achieve crop yield, and in a similar way to chemical fertilizer alone. The soil microbial network played an important role in maintaining soil fertility, soil nutrient conversion, and soil health [14,15]. In addition, soil microbial communities and nutrient cycling are usually affected by the interactions of agricultural management, such as fertilization [16,17] and crop straw return [18]. Nevertheless, the effects of the SLR-and-FR-combined treatment on the red soil microbial network is still unclear. However, their interrelation was much less studied. The conventional farming system overused chemical fertilizer; the main aim of fertilizer reduction is optimizing nutrient efficiency [19]. Therefore, it is crucial that we explore the impact of FR with SLR on the soil microbial network, soil properties, and their interrelation.

Red soil covers approximately 1.13 million km^2^ of Southern China, accounting for 11% of the national land area, which is the main and typical soil type in Southern China. Owing to long-term unreasonable fertilizer application and sugarcane leaf burden, red soil in Southern China has led to serious soil biodiversity degradation, which is generally characterized by low soil nutrients and crop yield, and high soil acidity [20]. Previous studies ignored the combined effects of SLR and FR on the soil microbial communities study on the interaction between soil properties and the soil bacterial network; this was less reported, especially the relationship on soil properties and the soil bacterial network, which was rarely reported. Hence, facilitating the research and popularization of SLR and FR on the red soil microbial network would bring remarkable social, economy, and ecological benefits.

The rapid development of high-throughput sequencing technology and bioinformation analysis has facilitated the soil microbial network research [21,22]. Based on the promotion and application of the third-generation high-throughput sequencing technology, further study is required in order to understand the relation of the soil bacterial network and soil properties under different SLR and FR practices; therefore, observations on the effects of the soil bacterial network under SLR and FR are necessary in subtropical red soil regions. In this study, we selected SLR and FR, the two main agricultural management methods in production, and collected red soil for the culture soil. We hypothesized that the different SLR and FR levels could significantly influence the effects of the soil bacterial network on soil properties and soil bacterial communities as the functional group structure. We have reported how SLR and FR affect the maize growth, yield, and soil properties in Liu et al. (2023) [23]. In this study, we primarily focused on the following hypotheses: (1) SLR and FR treatments would affect the soil bacterial network in red soil; and (2) there would be a relationship for the varying effects of the different SLR and FR treatment combinations on the soil properties and bacterial network.

## 2. Materials and Methods

### 2.1. Pot Experiment Setup and Materials Collection

A pot experiment was conducted at the plastic greenhouse (23°12′42″ N, 108°11′07″ E, 145 m altitude) in the Guangxi Vocational College of Water Resources and Electric Power, Nanning, Guangxi, China from February to July 2022. The tested soil was the typical red soil with a soil texture of clay in Southern China, which had been derived from the experimental field of the local campus on 19 February 2022. Before the tested soil was used, it was air-dried and crushed to pass through the 10.0 mm sieve, gravel was removed, and it was thoroughly mixed and stored. The initial soil properties were as follows: pH, 7.02; EC, 0.08 S/m; soil organic matter (SOM), 18.0 g/kg; total nitrogen (TN) content, 1.06 g/kg; total phosphorus (TP)content, 0.49 g/kg; total potassium (TK) content, 8.59 g/kg; alkali–hydrolyzable nitrogen (AN) content, 81.5 mg/kg; available phosphorus (AP) content, 2.0 mg/kg; and available potassium (AK) content, 52.0 mg/kg.

The sugarcane leaves were obtained from the sugarcane experiment station of the Guangxi Academy of Agricultural Sciences, Nanning, Guangxi, China on 19 October 2021. The properties of the sugarcane leaves were as follows: TN, 5.56 g/kg; TP, 1.24 g/kg; TK, 4.74 g/kg; total organic carbon content, 557.52 g/kg; and C/N ratio, 98.24. Before the use of sugarcane leaf residues, they were dried at 75 °C to a constant weight and artificially chopped to a length of 1.0–2.0 cm.

The hybrid maize (*Zea mays* L.) variety, with Yidan 629, was selected for the test. It is cultivated widely in Southern China. Maize was planted in black plastic buckets, with 30 kg of red soil filled in each bucket on an air-dried weight basis. The black plastic buckets, with 40.0 cm opening diameter, 32.0 cm base diameter, and 33.0 cm height, were used in the experiment. The tested maize was sown on March 16th and harvested on June 14th, with the whole growth period being 90 days. The tested maize seeds were planted with two seeds per bucket, 5.0 cm apart and 2.0 cm below the soil surface. All seeds sprouted completely on March 22nd. On March 25th, one seedling was removed from each pot, and only the stronger seedling was retained. The maize plants were irrigated artificially with running water to maintain the same irrigation water weight. Manual weeding and pest control for uniformity were performed during the whole growth period.

### 2.2. Experimental Design

The pot experiment was set up by using the randomized block design with SLR and FR as the experimental factors. Nine treatment combinations were established, using three different amounts of SLR [full SLR (FS), 120 g/pot; half SLR (HS), 60 g/pot; and no SLR (NS), with three FR levels including full fertilizer (FF), 4.50 g N/pot, 3.00 g P_2_O_5_/pot, and 4.50 g K_2_O/pot; half fertilizer (HF), 2.25 g N/pot, 1.50 g P_2_O_5_/pot, and 2.25 g K_2_O/pot; and NF], without nitrogen, phosphorous, and potassium added. Each treatment was replicated seven times, giving 63 pots in total. The SLR content was assumed as 9.0 t/hm^2^ using the average air-dried weight of sugarcane leaves in Southern China [24,25], and 2250 t/hm^2^ of the average surface soil weight [26]. Urea (N, 46.0%), calcium superphosphate (P_2_O_5_, 12.0%), and potassium chloride (K_2_O, 60.0%) were used as nitrogen, phosphorous, and potassium fertilizers, respectively. All fertilizers were applied with the analytical reagent, then mixed fully into the soil before the experiment. The detailed experimental design and the nutrient input of nine experimental treatments were shown in Liu et al. [23].

### 2.3. Soil Sampling

Soil sampling was carried out from March 16th to June 14th, with an interval of 15 days after maize was sown, with 7 sampling time points. Three soil cores were collected from each bucket at the same soil depths (0–10.0 cm). Fresh soil samples from each bucket were mixed thoroughly to form a composite sample. At every observation time point, 162 soil cores were collected and formed 54 composite soil samples. Fresh soil samples were divided into two sub-samples. One was placed in individual sterile plastic tubes and stored at −80 °C refrigerator before analyzing soil microbes. The others were air-dried for analysis of soil physicochemical properties. The assay results were based on the air-dried soil weight.

### 2.4. Soil Microbial Analysis

#### 2.4.1. DNA Extraction and PCR Amplification

Microbial DNA was extracted from soil samples using the E.Z.N.A^®^ Soil DNA Kit (Omega Bio-tek, Norcross, GA, USA) according to manufacturer’s protocols. The V1–V9 region of the bacteria 16S ribosomal RNA gene were amplified by PCR (95 °C for 2 min, followed by 27 cycles at 95 °C for 30 s, 55 °C for 30 s, and 72 °C for 60 s, and a final extension at 72 °C for 5 min) using primers 27F 5′-AGRGTTYGATYMTGGCTCAG-3′ and 1492R 5′-RGYTACCTTGTTACGACTT-3′, where barcode is an eight-base sequence unique to each sample. PCR reactions were performed in triplicate 20 μL mixture containing 4 μL of 5×FastPfu Buffer, 2 μL of 2.5 mM dNTPs, 0.8 μL of each primer (5 μM), 0.4 μL of FastPfu Polymerase, and 10 ng of template DNA. Amplicons were extracted from 2% agarose gels and purified using the AxyPrep DNA Gel Extraction Kit (Axygen Biosciences, Union City, CA, USA) according to the manufacturer’s instructions.

#### 2.4.2. Library Construction and Sequencing

SMRTbell libraries were prepared from the amplified DNA by blunt ligation according to the manufacturer’s instructions (Pacific Biosciences, Menlo Park, CA, USA). Purified SMRTbell libraries from the Zymo and HMP mock communities were sequenced on dedicated PacBio Sequel cells using the S/P1-C1.2 sequencing chemistry. Purified SMRTbell libraries from the pooled and barcoded samples were sequenced on a single PacBio Sequel cell. Replicate 1 of the samples was sequenced using the S/P2-C2/5.0 sequencing chemistry and replicate 2 of the samples was sequenced with a pre-release version of the S/P3-C3/5.0 sequencing chemistry. All amplicon sequencing was performed by Shanghai Biozeron Biotechnology Co., Ltd. (Shanghai, China).

#### 2.4.3. Processing of Sequencing Data

PacBio raw reads were processed using the SMRT Link Analysis software version 9.0 to obtain de-multiplexed circular consensus sequence (CCS) reads with the following settings: minimum number of passes = 3, and minimum predicted accuracy = 0.99. Raw reads were processed through SMRT Portal to filter sequences for length (>1200 bp) and quality. Sequences were further filtered by removing barcode and primer sequences.

OTUs were clustered with 98.65% similarity cutoff using UPARSE (version 10, http://drive5.com/uparse/ (accessed on 16 January 2023). In the case of multi-grouping, to understand the OTU differences among different groupings, a flower diagram analysis was carried out based on OTU information. The phylogenetic affiliation of each 16S rRNA gene sequence was analyzed by uclust algorithm (https://github.com/topics/uclust (accessed on 21 January 2023) against the Silva (SSU138.1) 16S rRNA database (http://www.arb-silva.de (accessed on 25 January 2023) using confidence threshold of 80% [27].

#### 2.4.4. Alpha and Beta Diversity

The rarefaction analysis based on Mothur v. 1.21.1 [28] was conducted to reveal the diversity indices, including the Chao1, Simson, Shannon et al. The beta diversity analysis was performed using UniFrac [29] to compare the results of the principal component analysis (PCA) using the community ecology package, R-forge (Vegan 2.0 package was used to generate a PCA figure). One-way analysis of variance (ANOVA) tests were performed to assess the statistically significant difference of diversity indices between samples. Differences were considered significant at *p* < 0.05. Flower diagrams were drawn using Perl (v5.26.3) software to analyze shared and unique OTUs of the bacterial communities in different treatment.

#### 2.4.5. LEfSe Analysis

For identification of biomarkers for highly dimensional colonic bacteria, LEfSe (linear discriminant analysis effect size) analysis was carried [30]. Kruskal–Wallis sum-rank test was performed to examine the changes and dissimilarities among classes, followed by LDA analysis to determine the size effect of each distinctively abundant taxa [31].

### 2.5. Data Analysis

Multivariate analysis of variance (ANOVA) was performed following the general linear model univariate procedure using SPSS Statistics 27.0 software (IBM, New York, NY, USA). Significant differences between different treatments were calculated using the least significant difference (LSD) method (*p* < 0.05). The correlation between soil bacterial network and soil proprieties was assessed via the two-tailed significance test using the Pearson coefficient.

## 3. Results

### 3.1. The Composition and Diversity of Soil Bacterial Species

A total of 114,011 operational taxonomic units (OTUs) were obtained from the soil bacterial community analysis, which were assigned to 46 phyla, 83 classes, 180 orders, 407 families, 1387 genera, and 5099 species. In this study, taxonomic profiling showed that the bacterial community was dominated by the genus level (Figure 1).

The microbial diversity analysis indicated that the microbiota included nine major phyla: *Acidobacteria*, *Actinobacteria*, *Bacteroidetes*, *Chloroflexi*, *Gemmatimonadetes*, *Planctomycetes*, *Proteobacteria*, *Unclassified*, and *Verrucomicrobia* were the dominant bacterial phyla across all treatments. Additionally, *Actinobacteria* was the most abundant phyla in all samples.

The soil bacterial phyla with their genus level associated with the nine treatment groups at seven sampling time points was shown in the stacking diagram (Figure 2). Bacterial sequences accounted for almost 100% of the reads and the top 20 phyla had been shown with a different color in Figure 2. The effect of SLR on soil bacteria was a significant effect on the *Unclassified* at 15 and 90 days (*p* < 0.05), respectively. The effect of FR on soil bacteria had a significant effect on the *Unclassified* at 15 days, and the *Ramlibacter* at 30 and 60 days (*p* < 0.05), respectively. The SLR-and-FR-combined treatments had no significant effect on the proportion of three soil bacteria. Three predominant phyla, viz. *Ramlibacter*, *Unclassified*, and *Luteitalca,* were used to compare the changes in different trial treatments. Except at 45 days, the significant difference in *Ramlibacter*’s proportion among nine treatment combinations was found at other six observation time points (*p* < 0.05). A significant difference in *Luteitalca*’s proportion and *Unclassified*’s proportion was found to have no difference, and, at 15 days, between nine treatments at seven sampling time points, respectively. On the first day, the *Ramlibacter* proportion of FSHF was higher by 68.95% and 14.25% than that of NSNF and FSNF, respectively. On the fifteenth day, the *Ramlibacter* proportion of HSNF and NSFF was lower by 40.92% and 46.87% than that of NSNF. The *Ramlibacter* proportion of HSFF and HSHF was higher by 12.92% and 58.40% than that of HSNF, respectively. The *Unclassified* proportion of HSHF was higher by 12.84% and15.70% than that of NSNF and HSNF, respectively. On the thirtieth day, the *Ramlibacter* proportion of FSFF, FSHF, FSNF, HSFF, HSHF, HSNF, NSFF, and NSHF was lower than that of NSNF (CK). Under the same SLR conditions (FS, HS, and NS), the *Ramlibacter* proportion of FF and HF was lower than that of NF. The *Unclassified* and *Luteitalca* proportion was exactly the opposite. On the 45th day, the changes in the *Ramlibacter* proportion were similar with the 30-days’ trend. The unclassified proportion of FSFF and HSFF was higher by 36.79% and 9.81% than that of NFNF; meanwhile, the unclassified proportion of FSNF was lower by 32.30% than that of FSFF. The *Luteitalea* proportion of FSFF, FSHF, FSNF, HSFF, HSHF, HSNF, NSFF, and NSHF was lower than that of NSNF(CK). On the sixtieth day, the changes in the *Ramlibacter* proportion was similar with the 30- and 45-days’ trend. The *Unclassified* proportion of FSFF and HSNF was higher by 33.55% and 21.29% than that of NSNF, respectively. The *Unclassified* proportion of FSFF and FSHF was higher by 13.74% and 10.81% than that of FSNF, respectively. The changes in the *Luteitalea* proportion was similar with the 45 days’ trend; meanwhile, the *Luteitalea* of FSFF, HSFF, and NSFF was lower by 19.41%, 21.79%, and 20.86% than that of FSNF, HSNF, and NSNF, respectively. On the 75th day, the *Ramlibacter* proportion of FSFF, FSHF, FSNF, HSFF, HSHF, HSNF, NSFF, and NSHF was lower than that of NSNF(CK); the changes trend was similar with the 30th, 45th, and 60th day. The *Unclassified* proportion of HSNF was higher by 14.16% than that of NSNF(CK). The *Unclassified* proportion of FSHF was higher by 13.50% than that of FSNF(CK). Under the same SLR condition, the *Luteitalea* proportion of FF and HS was higher than that of NF. On the 90th day, the *Ramlibacter* proportion of NSHF was higher by 20.43% than that of NSNF. The *Unclassified* proportion of NSNF (CK) was lower by 46.33% than that of FSFF. The *Luteitalea* proportion of HSFF and HSHF was higher by 11.21% and 10.28% than that of NSNF, respectively.

The species distribution in different treatment combinations were shown to have some similarity and specificity. A flower plot showing the unique and shared OTUs of the bacterial communities of nine treatments at seven sampling time points (Figure 3). The distribution of sequences demonstrated that each treatment appeared to involve a distinct microbial population, respectively. The results indicated that the number of the unique and shared OTUs was related with the maize growth stage.

The 33 unique OTUs were found on the first day after sowing, then reached a higher value of 62 on the 15th day (Figure 4), then decreased constantly on the 30th and 45th day. But, on the 60th day, the core OTUs reached a peak value of 88, then decreased continually from the 60th to 90th day, the minimum value of 27 on the 90 day. The experimental results showed that the number of core OTUs increased when the maize grew rapidly (15 days and 60 days, and jointing and filling stage, respectively), and decreased when the maize growth slowed and aged (the 0th day and from the 60th to the 90th day, and the seeding and ripening stage, respectively). This implied that the number of core OTUs of different treatment combinations were relevant to the maize growth stage to some extent.

### 3.2. The Diversity Analysis of Soil Bacteria Network

For the differences evaluated in the composition of the soil samples subjected to different treatments, principal component analysis (PCA) was conducted, which was based on the Bray–Curtis matrix. Treatments with a similar bacterial community composition were clustered together, which were regulated by SLR and FR. As shown in Figure 5, the first two principal components (PC_1_ and PC_2_) together explained 9.67%, 10.87%, 10.45%, 9.5%, 10.08%, 10.00%, and 8.87% of the variation in the soil bacterial community on the 0th, 15th, 30th, 45th, 60th, 75th, and 90th day, respectively. The PCA ordinations showed that soil microbial communities were significantly separated by sugarcane leaf return (FS and HS) and no sugarcane leaf return (NS) treatments on the 15th day. At the same time, soil bacterial communities were divided into two groups, whereby sugarcane leaf return (FS and HS) treatments (FSFF, FSHF, FSNF, HSNF, and HSFF) represented one group, and no sugarcane leaf return (NS) treatments (NSHF and NSNF) were clustered into the other group. The response of the soil bacterial network to SLR in different fertilization levels was indistinct when observed for soil bacterial on the other sampling time point. The composition of the soil bacterial network did not show a significant difference between the different treatment combinations; FR had less of an effect on changing soil bacterial network composition, compared to SLR. Therefore, in the following soil bacterial communities’ analysis, the effects of SLR and FR were separately applied.

The diverse microbial communities and different phylogenetic OTUs in the soil samples were revealed from high-quality sequencing reads. Alpha diversity analyses could describe the abundance and diversity of the soil bacterial network. The alpha diversity metrics include species richness (Chao1, Shannon, Simpson, et al.) [32]. The Chao1 index was the species-richness index of the soil bacteria community, which was used to assess the OTU number. We used the Chao1 index to assess the alpha diversity difference of nine treatment combinations in seven sampling time points (Figure 6). On the 0th, 15th, 30th, 45th, 60th, 75th, and 90th day, the variation range of the Chao1 index was 5000–9000, 3000–4500, 4500–6500, 3000–6000, 3500–5500, 3500–6500, and 2500–4500, respectively. On the 0th day, under FS treatments, there were no significant differences in the Chao1 index in the average diversity of the FF, HF, and NF groups. In addition, along with the HS condition, the Chao1 index value of FF and HF was higher than that of NF, respectively. Similar results were found under the NS condition. The variation range of the Chao1 index in HSNF was bigger than that under NSNF treatment. On the 15th day, the Chao1 index value of FSFF, FSHF, was greater than that in FSNF treatment, but was no significance. But the variation range of the Chao1 index of HSNF and NSNF was significantly bigger than the other treatments. On the 30th day, the Chao1 index value of FSFF was significantly greater than that of FSHF and FSNF; it was just that the Chao1 index value of HSFF was higher than that of FSNF. In addition, similar results were found when using the Chao1 index to compare the HSFF, HSHF, and HSNF groups. Under the NS condition, the Chao1 index of NSFF was significantly greater than that of NSNF. On the 45th day, the Chao1 index value of FSNF was greater than that of FSFF, but it was lower than that of FSHF. Under the HS condition, the value of the Chao1 index was fairly close between HSFF, HSHF, and HSNF. Meanwhile, the Chao1 index value of NSNF was significantly greater than that of NSFF and NSHF. On the 60th day, under the FS condition, the Chao1 index value of FSNF was significantly bigger than that of FSFF and FSHF, respectively. Similar results were found between the HSFF, HSHF, and HSNF groups. But, under the NS condition, the Chao1 index value of NSNF was lower than that of NSFF and NSHF. On the 75th day, under the FS condition, the Chao1 index value of FSHF was significantly greater than that of FSFF and FSNF. However, the Chao1 index value of HSFF was obviously greater than that of HSHF and HSNF. Under the NS condition, the Chao1 index value of NSFF and NSHF was bigger than that of NSFF. On the 90th day, the Chao1 index value of FSFF was significantly lower than that of FSHF and FSNF. The results of HSFF, HSHF, and HSNF were close. In addition, similar treads were found between the NSFF, NSHF, and NSNF groups.

As shown in Figure 7, on the 0th day, in the linear discriminant analysis (LDA) of the soil bacterial network, the seven trial treatments (FSHF, HSFF, HSHF, HSNF, NSFF, NSHF, and NSFF) were not found to have a significant difference in soil bacterial discriminant taxonomic nodes. On the 15th day, there was a significant difference of the four trial treatments (FSFF, FSNF, HSFF, and NSFF) in soil bacterial classification levels. The *chthonomaonadate* in NSFF was significantly difference at the class, order, and family levels, respectively; meanwhile, *Isosphacles* of NSFF was significantly different at the order and family levels. A significant difference of *Chloroflexi* at the phylum level was found in HSFF (*p* < 0.05). On the 30th day, only three treatments (HSFF, HSHF, and NSFF) of the soil bacterial network was found to be significantly statistically and biologically differentiated. The *Herpetosiphonace* in NSFF was significantly different at the order and family level. The *Herpetosiphonacease* and *Herpetosiphonales* in HSHF was significantly different at the order and family level. The *Endterobacteriaceae* in NSFF was significantly different at the family level (*p* < 0.05), respectively. The results of the LDA of the soil bacteria network on the 45th day showed that six treatments (FSFF, FSHF, HSHF, HSNF, NSFF, and NSHF) were found to be different in soil bacterial species taxonomy; the difference of HSHF, NSFF, and NSHF on the soil bacterial network was significantly different (*p* < 0.05). The *Herpetosiphonace* and *Bifidobacteriales* in HSHF was significantly different at the family and genus level, respectively; meanwhile, the *Burkholderia* and *Betaproteoba* in NSNF was found the be significantly different at the class and order level (*p* < 0.05). The results were found to be significantly different in the LDA on the 60th day (*p* < 0.05). The *Acidobacteria* in NSNF was significantly different at the order and family level. The *Thermoanacerobaculia* in HSFF was found to be significantly different at the class, order, and family level. The *Anaerolineac* in HSHF was significantly different at the class, order, and family level. On the 75th day, the significant difference in soil bacterial species taxonomy was found in six treatment combinations (FSFF, FSHF, FSNF, HSNF, NSFF, and NSNF). The *Bacteroidales*, *Bacteroidia*, and *Dcinococci* in FSFF and the *norank*, *CandidatusPeribacterales*, and *CandidatusPeribaceria* in HSNF were found to be significantly different at the class, order, and family level. On the 90th day, the difference was found obviously in seven treatments (FSFF, FSHF, FSNF, HSHF, HSHF, HSNF, NSFF, and NSNF); the *Pyrinomonadaceae*, *Blastocatellales*, and *Blastocatellia* in HSHF and the *Candidatus Chazhemtobateraceae* and *norank* in NSNF were found to be significantly different at the class, order, and family level. Meanwhile, the *Lacipirellulaccae* and *Azonexaceae* in FSNF was significantly different at the family level.

### 3.3. The Correlation Analysis of the Soil Bacterial Network and Soil Properties

The correlation analysis of the soil bacteria network and soil properties was performed to analyze (see data in Liu et al. [23]) to further investigate the interaction between them (Figure 8). The heatmap showed the relative abundance of selected soil microbial OTUs at the genus level associated to the nine soil physicochemical indices (peak), in different SLR and FR treatments. *Flavisolibacter* and *Niastella* was negatively correlated with pH, significantly (*p* < 0.05). Meanwhile, there was a significantly positive correlation between five soil bacterial species (*Bellilinea*, *Unclassified*, *Gemmata*, *Longilinea*, and *Fimbriiglobus*) and pH (*p* < 0.05). *Flavisolibacter*, *Massilia*, *Lysobacter*, *Sphingomonas*, and *Sphingomicrobium* had a significantly positive correlation with EC, but there was a significantly negative correlation between *Gemmata*, *Pirellula*, *Longilinea*, *Fimbriiglobus*, and EC. *Flavisolibacter*, *Massilia*, *Lysobacter*, *Sphingomonas*, *Pseudoduganella*, *Sphingomicrobium,* and *Ohtackwangia* were positively correlated with TN; however, there was a significantly negative correlation between *Unclassified* and TN. Only *Unclassified* had a significantly negative correlation with TP. *Flavisolibacter*, *Massilia*, *Sphingomonas*, *Pseudoduganella*, *Ramlibacter*, *Sedminibacterium*, and *Niastella* were negatively correlated with TK; however, *Luteitalea*, *Unclassified*, and *Pirellula* were just the opposite. There was a significantly positive correlation between *Massilia*, *Lysobacter*, *Pseudoduganella*, *Paraflavitalea*, and AN; *Unclassified*, *Gemmata*, *Pirellula*, *Longilinea*, and *Fimbriiglobus* had a significantly negative correlation with AN. *Bellilinea*, *Gemmata*, and *Fimbriiglobus* had a significantly negative correlation with AP, but *Unclassified* was just the opposite. We noted that *Unclassified*, *Gemmata*, *Pirellula*, *Longilinea*, and *Fimbriiglobus* had a significantly negative correlation with AK. There was a significantly opposite correlation between *Lysobacter*, *Pseudoduganella*, and SOC; however, *Bradyrhizobium*, *Bellilinea*, and *Gemmata* had a significantly negative correlation with SOC.

## 4. Discussions

Soil is a great home to many microorganisms that play important roles in soil nutrient conversion, energy transformation, and the formation of humus [33,34]. Soil microbes are essential links between above- and below-ground ecosystems, which maintain soil ecosystems and the soil sustainable productivity [35]. The relationship between soil and the bacterial network has made the bacterial community an essential indicator of the soil quality and environment. The structure and diversity of soil microbial communities are sensitive to external environmental factors, such as fertilization [16,17] and straw return [36,37]. The study of SLR and FR on the red soil bacterial network was rarely reported. The sugarcane leaf was mainly returned to the sugarcane field and was less often returned to other crops. Soil microbes could have been affected by crop straw and fertilization; meanwhile, the soil can affect soil microorganisms by changing the quantity and quality of soils in crop straw return or fertilization, and the soils can serve as carbon and nitrogen sources and other nutrients for the microbes and crop.

### 4.1. Interaction of SLR and FR on Soil Bacteria Community and Diversity

#### 4.1.1. Changes in Soil Bacterial Community’s Major Types during Maize Growth

Despite the differences in SLR and FR, the bacterial community structure in the red soil at the genus level was similar among nine treatment combinations on seven sampling time points. Similar soil actuarial communities were found in the tropical rainforest in Malaysia and the temperate forest in Japan, in which *Acidobacteria*, *Actinobacteria*, and *Proteobacteria* occupied the top phyla [38]. Similar soil bacterial community compositions were also found in vineyard soil [39], riverine wetland [40], copper mines [41], and a shallow lake [42]. Our results indicated that the SLR and FR methods did not transform the main taxonomic structure at the soil bacterial genus level. According to the results of the relative abundance of soil bacterial communities at the genus level (Figure 2), SLR and FR increased the complexity of the soil bacterial network in red soil. However, FS was clustered together with HS, which suggested that SLR may have a greater impact on the soil bacterial network than FR. Among all combinations, differences between groups were greater than intra-group deference.

#### 4.1.2. Maize Growth Stage Affected Soil Bacterial Community Structure and Diversity

Ecological rules for the microbial community structure and assembly have been gradually applied to the diversity analysis for various soil microorganisms [43]. Most studies have shown that changes in the microbial communities in soil are closely related to environmental factors and seasonal changes. Using co-occurrence analysis, Fang et al. [44] found that microbial communities have a more complex and stable structure in summer compared with winter. Zhao et al. [45] found that growth periods significantly changed the bacterial and fungal sub-communities’ structure. Xu et al. [46] indicated that the maize growth stage, rather than the maize cultivar, was the main factor causing changes in soil bacterial communities. In the current study, the difference from SLR and FR had less of an effect on soil bacterial communities compared with the maize growth stage. Based on the analysis results of the flower plot (Figure 3 and Figure 4), the number of core OTUs was still at a high level on the 15th and 60th day (the jointing and tasseling stage). The results were just the reverse on the start and the 90th day. Similar to alpha diversity, LEfSe thus identified several key phylotypes from the phylum to family level that were statistically different between the nine treatments. Although there was no substantial change in soil bacterial communities among sampling time points between SLR and FR treatments, LEfSe identified some key phylotypes that were statistically different among treatments, according to the analysis results of LDA, and found that the number of trial treatments with a difference and the number of soil bacterial species with a difference were closely associated with the maize growth stage (Figure 9). In the early and latter stages of maize growth, the frequency of trial treatments with a difference and soil bacteria with a difference was rare (0th, 15th, 30th, and 75th day). However, the results were the opposite at the stage of maize peak growth (60 day). This finding is consistent with the results of most previous studies.

The results of the Chao1 index showed that the abundance and diversity of the soil bacterial network were also associated with the maize growth stage. The change trend of the Chao1 index was similar with the LDA results. SLR and FR could affect the abundance and diversity of the soil bacterial network to some extent. Under the same fertilization level, the Chao1 index value of the three sugarcane leaf return amounts was always as follows: FS > HS > NS. Meanwhile, the Chao1 index value of the three fertilization levels was in the order of FF > HF > NF.

Bacteria are core components of the soil microbiota and perform critical ecological functions in terrestrial ecosystems, as they are involved in most nutrient transformations, energy flow, or organic matter turnover, and thereby affect above-ground crop growth and yield [47]. Changes in abiotic and biotic environmental variables are usually responsible for shaping the soil bacterial network structure [7]. For example, SLR or FR could change the soil properties and alter the osmotic strength of the soil solution; thereby, the changes also influence soil microbial activities including the soil enzyme activity, soil microbial biomass, and soil bacterial abundance and diversity, similar to the effects of crop growth. In the present study, we characterized the shifting patterns of soil bacterial network under different SLR and FR treatments. Consistent with our hypothesis, bacterial communities and their interactions were affected by SLR and FR in red soil of Southern China. We also identified some keystone bacteria species, which might be responsible for maintaining the stability and functions of ecosystems in the semitropical red soil zone.

#### 4.1.3. SLR Significantly Influences Red Soil Bacterial Community Compositions

The straw return technique is always considered an effective measure for improving soil quality and maintaining soil microorganisms [37]. When crop residues are incorporated into the soil, soil microbial growth can be stimulated and further promote straw decomposition. Microbial-mediated straw decomposition is, in essence, a process of nutrient release, organic C mineralization, and soil organic C balance [48,49], resulting in an increase in inorganic nutrients and organic matter in the soils [50]. However, straw decomposition might be dominated by different soil microbes with specific functions at different decay stages [51]. Zhao et al. [52] reported that low rates (≤4500 kg/ha) of 30-year straw incorporation did not have an effect, but high rates (9000 kg/ha) of straws largely shifted the soil microbial community structure. Jiao et al. [53] investigated the black soil microbial communities with different straw returning and nitrogen fertilizer reduction treatment combinations; their results showed that the soil bacterial communities were consistently dominated by *Acidobacteria*, *Proteobacteria*, *Actinobacteria*, and *Chloflexi*. In the present study, the dominant soil bacterial species were *Acidobacteria*, *Actinobacteria*, *Bacteroidetes*, and *Chloroflexi* (Figure 1), which was similar to Jiao et al. [53].

#### 4.1.4. FR Partly Influences Red Soil Bacterial Network Compositions

Routine applications of fertilizer are an important agricultural practice for improving the nutrient availability of the soil and increasing crop yield, ultimately [54]. Soil bacterial networks are important for soil fertility due to the role they play in organic matter decomposition, nutrient cycling, and the formation of organic complexion agents and the maintenance of soil structure. It has generally been shown that mineral fertilizers can directly stimulate the growth of specific bacteria populations by the nutrient supply added [16], leading to an increase in total bacterial numbers [55,56], improving the soil bacterial activity [57], and determining a switch in soil bacterial diversity. Hence, soil bacterial networks are strongly affected by the application of fertilization [17]. In a long-term field experiment, Francioli et al. [58] found a greater bacterial diversity and richness in soils fertilized with farmyard manure than in soils receiving mineral fertilizer. Other field studies support these findings. For example, Li et al. [59] found that the annual application of organic fertilizer increases the bacterial community biomass and diversity, while Hartmann et al. [60] reported an increased richness and decreased evenness of bacterial communities in soils under organic management, which was primarily attributed to the use and quality of the organic inputs. Wang et al. [17] reported that the combination of chemical fertilizer with manure could improve the soil bacterial abundance significantly, and, therefore, reduce the adverse effects and potential risks induced by excessive fertilizer application. Geisseler et al. [61] suggested that chemical fertilization did not consistently select for specific microbial groups; however, it affects the microbial community composition through the nutrient supply added and the changes in soil properties. The results were similar with Dincă [16]. In the present study, we found the FR could not affect the soil bacterial community based on the results of PCA (Figure 5). At the same time, the soil bacterial diversity and richness were greater in soils fertilized with SLR and FR than in soils receiving SLR without FR, which was similar to Francioli [58].

### 4.2. Correlations between Soil Bacterial and Soil Properties

In the study, we indicated that the interrelation between the top 20 soil bacteria and nine soil properties indices was more fluctuation. The soil pH was significantly positively correlated with five soil bacteria species, but negatively correlated with two soil bacteria species. The interrelation between EC, TN, TK, AN, and SOC, and the top 20 soil bacteria was similar to that of soil pH. Simultaneously, there was a significant negative correlation between TP and AK with one to five soil bacteria species (*p* < 0.05). The analysis results implied that the species and abundance of the soil bacterial network were related to the soil pH. Increasing the contents of TK, AN, SOM, and EC in red soil could stimulate the richness and correlation of soil bacterial species.

Numerous study results had confirmed that the soil environmental factors played a dominant role in constructing the structure of the soil bacterial network [56,57]. *Chloroflexi* (bacterial taxa) is relatively abundant in agricultural soils and could ferment sugars and polysaccharide into hydrogen and organic acids, which can promote the degradation of plant residues, fix the nitrite, and reduce the nitrate content [62]. Similar to *Proteobacteria*, *Acidobacteria* play an important role in carbon and nitrogen metabolism, and they are both highly abundant in agricultural soil environments [63]. *Bacteroidetes* are mainly responsible for nitrification processes, including autotrophic metabolism and the subsequent nitrite oxidation [64]. The soil properties indices were positively correlated with the soil bacterial network, partly; meanwhile, they were negatively correlated with the other soil bacteria species, except TP and AK. For the correlation analysis results on the soil bacterial network, the correlation between the soil bacteria and soil properties indices was relatively scattered. Given the large differences in experimental factors in the short-term pot test, more work on characterizing the interactions of SLR-FR application is needed in a long-term field experiment.

## 5. Conclusions

In summary, soil microbial networks were more active in response to FR than SLR, whereas the soil bacterial network was more sensitive to soil differences. The significant structural reorganization of the soil bacterial network between SLR and FR treatments was illustrated by principal component analysis, and the composition of the bacterial community was significantly different among different SLR and FR treatments. The sugarcane leaf and fertilizer reduction had significantly changed the structure and composition of the red soil bacterial community and increased the complexity of soil bacteria. Relative to NSNF, the combined SLR and FR application significantly increased the soil bacterial activity. Sugarcane leaf return and fertilizer reduction had increased the complexity of soil bacteria significantly; the effect of SLR on the soil bacterial community and diversity was higher than that of FR. The soil bacterial community and diversity were significantly affected by the maize growth stage. SLR and FR application could influence soil properties by altering soil bacterial communities. *Acidobacteria*, *Actinobacteria*, and *Bacteroidetes* were the dominant bacterial phyla across all treatments. The contents of TK, AN, and SOM, and EC were closely relevant to the soil bacterial network in Southern China red soil. Future studies need to focus more on the SLR and FR combination to understand how SLR/FR interactively affect below-ground ecological processes in agricultural ecosystems during long-term field conditions.

## Figures and Tables

**Figure 1 microorganisms-12-01788-f001:**
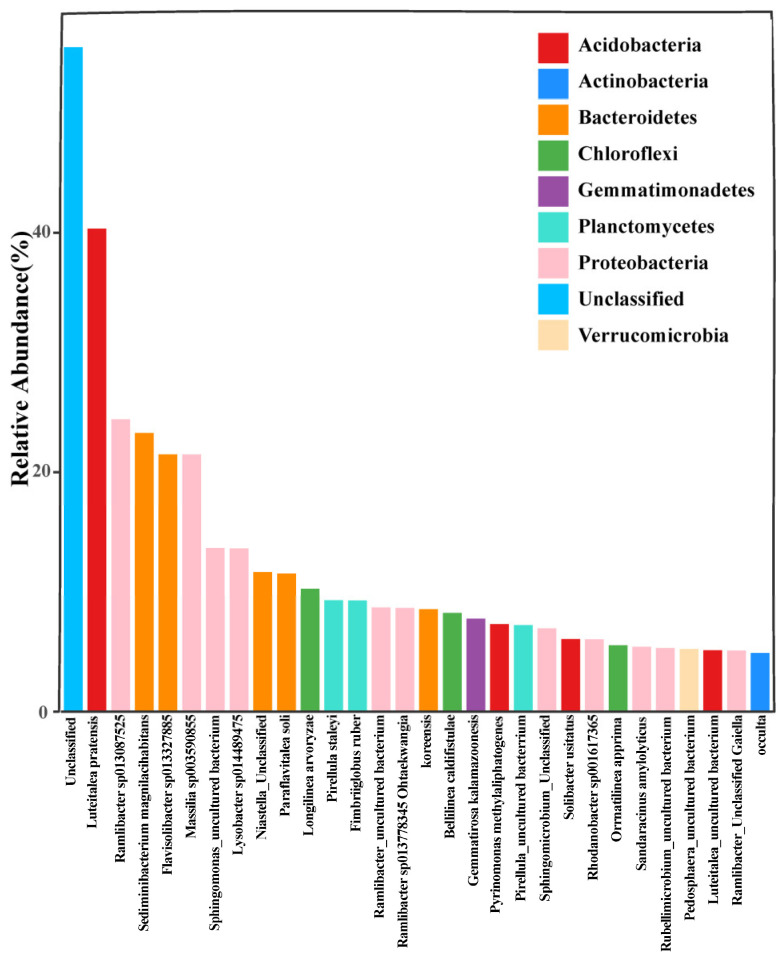
Dominant species microbial community component structure of all test treatments at genus level.

**Figure 2 microorganisms-12-01788-f002:**
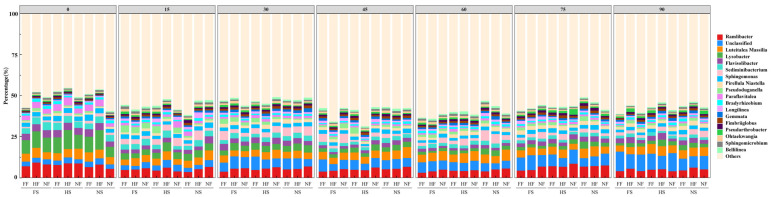
Stacking histograms of the relative abundance of soil bacterial communities at the genus level in 7 sampling time points. FS: full sugarcane leaf return, HS: half sugarcane leaf return, NS: No sugarcane leaf return; FF: full fertilizer, HF: half fertilizer, NF: no fertilizer.

**Figure 3 microorganisms-12-01788-f003:**
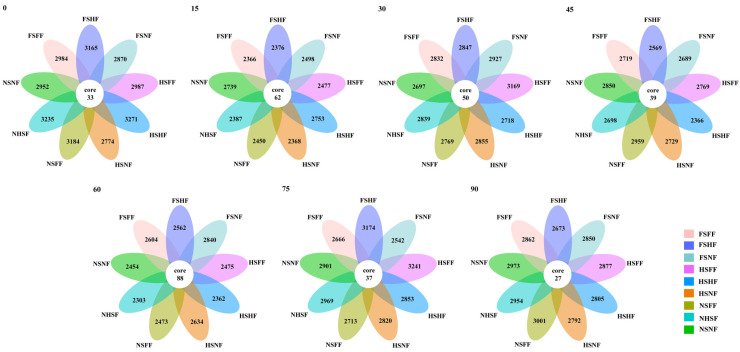
Flower plot among different SLR and FR treatment combinations based on OTUs in 7 sampling time points.

**Figure 4 microorganisms-12-01788-f004:**
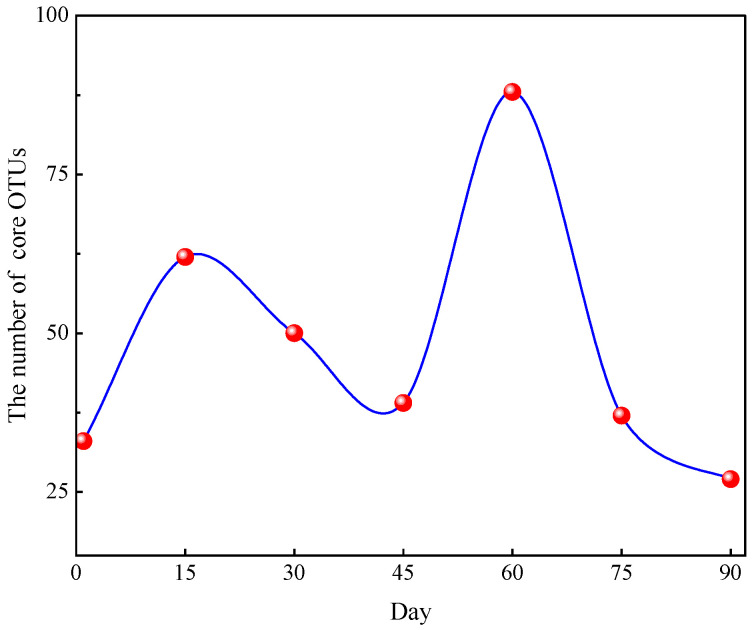
The changes in core OTU number in 7 sampling time point.

**Figure 5 microorganisms-12-01788-f005:**
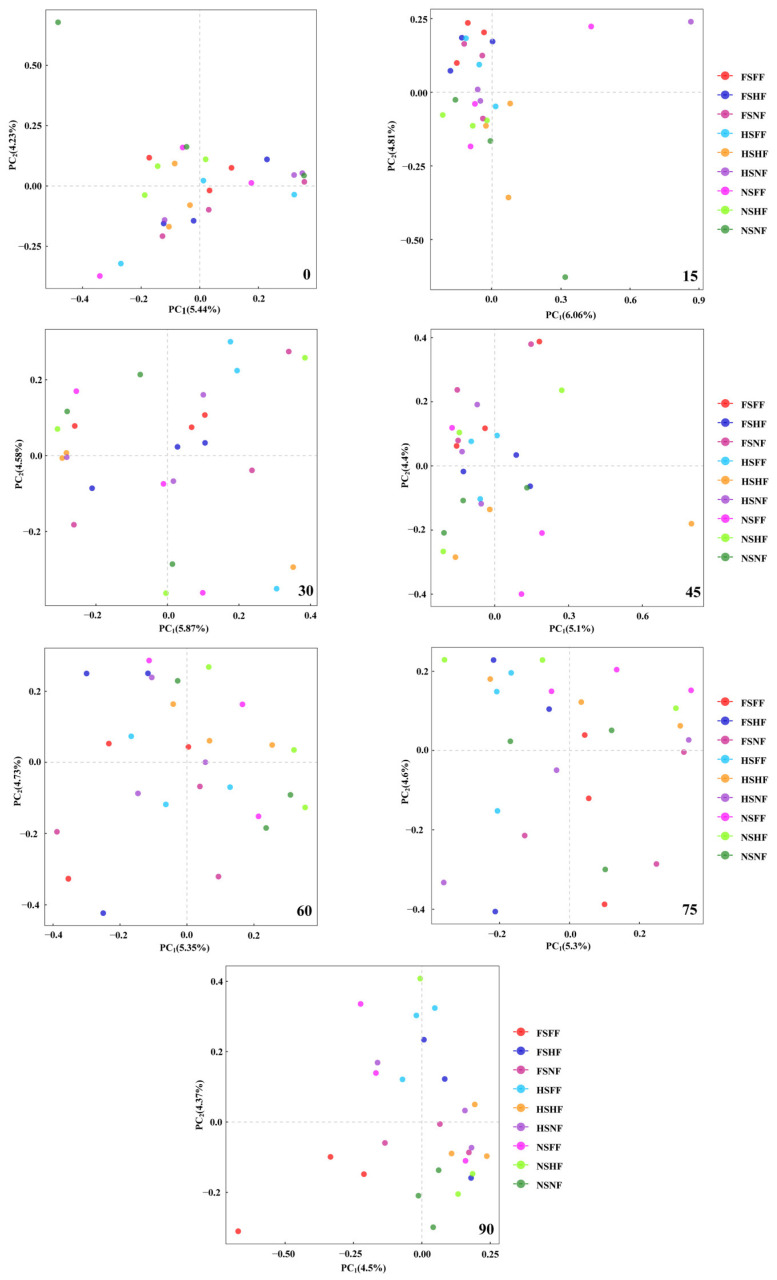
PCA of soil bacterial network of different SLR and FR treatment combinations in 7 sampling time point.

**Figure 6 microorganisms-12-01788-f006:**
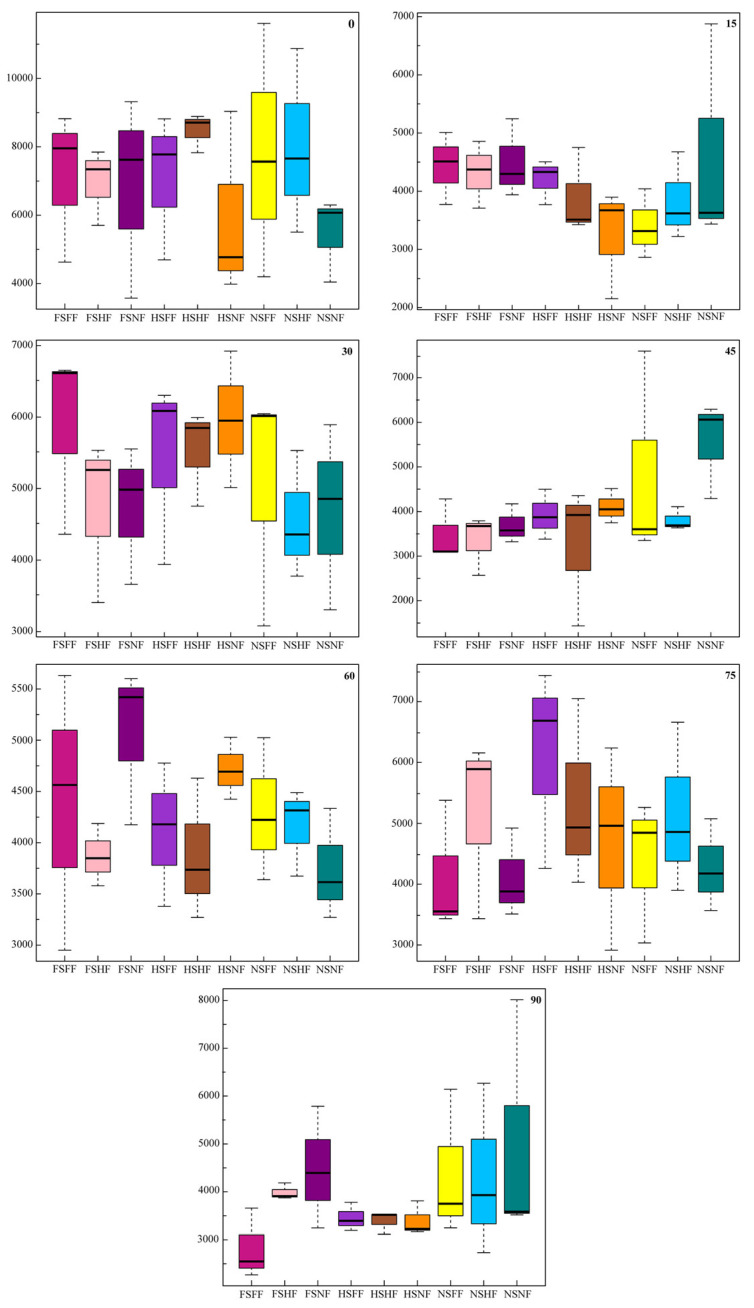
Chao1 index for alpha diversity of different SLR and FR treatment combinations in 7 sampling time point.

**Figure 7 microorganisms-12-01788-f007:**
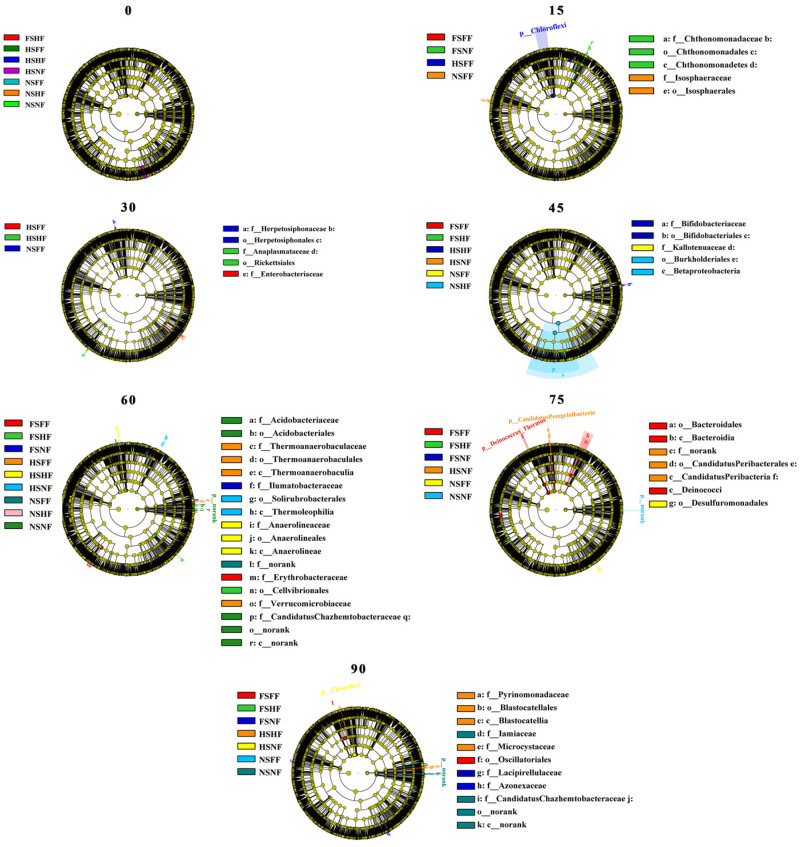
Linear discriminant analysis (LDA) effect size (LEfSe) taxonomic cladogram highlights the bacterial biomarkers that statistically and biologically differentiate groups in 7 observation time points. Significant and non-significant discriminant taxonomic nodes are colored. Each circle’s diameter reflects the abundance of that taxon in the community. The five rings of the cladogram stand for domain (innermost), phylum, class, order, and family. FSFF: full sugarcane leaf return + full fertilizer; FSHF: half sugarcane leaf return + half fertilizer; FSNF: full sugarcane leaf return + no fertilizer; HSFF: half sugarcane leaf return + full fertilizer, HSHF: half sugarcane leaf return + half fertilizer; HSNF: half sugarcane leaf return + no fertilizer; NSFF: no sugarcane leaf return + full fertilizer; NSHF: no sugarcane leaf return + half fertilizer; NSNF: no sugarcane leaf return + no fertilizer.

**Figure 8 microorganisms-12-01788-f008:**
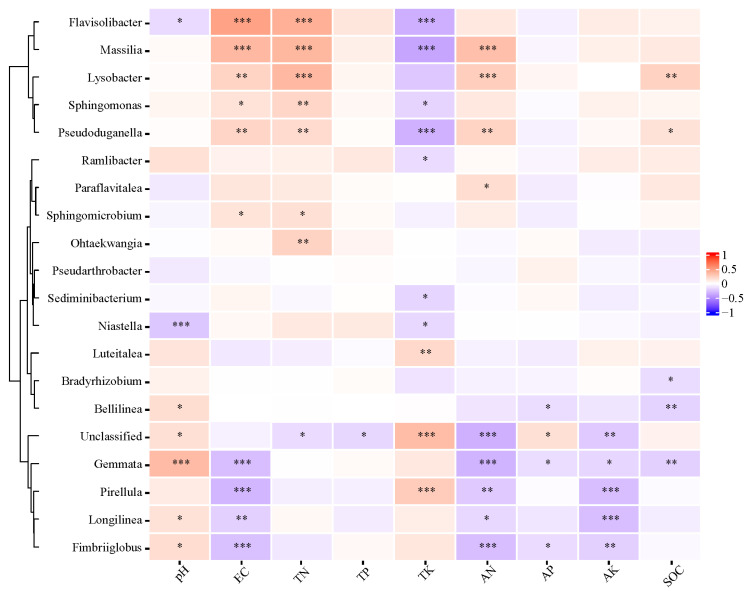
Heatmap of the correlation between top 20 bacterial microorganisms (genus level) and soil property indices based on Spearman correlation analysis. *, **, and *** indicate significant relationship at level of *p* < 0.05, *p* < 0.01, and *p* < 0.001, respectively. pH: soil potential of hydrogen; EC: soil electrical conductance; TN: soil total nitrogen content; TP: soil total phosphorus content; TK: soil total potassium content; AN: soil alkali–hydrolyzable nitrogen content; AP: soil available phosphorus content; AK: soil available potassium content; SOC: soil organic matter content.

**Figure 9 microorganisms-12-01788-f009:**
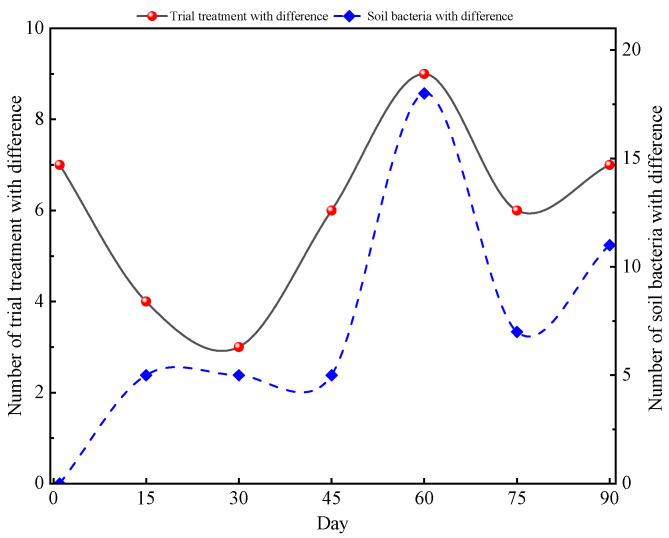
Changes in the number of trial treatment with difference and soil bacteria with difference during the test.

## Data Availability

All sequences in this work as well as metadata have been obtained from the Biozeron (http://www.biozeron.com/ accessed on 16 January 2023).

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
