# Peer review of "Interactions between Sugarcane Leaf Return and Fertilizer Reduction in Soil Bacterial Network in Southern China Red Soil"

_microorganisms, 2024, doi:10.3390/microorganisms12091788_

Round 1

Reviewer 1 Report

Comments and Suggestions for Authors

Dear Authors, the manuscript entitled " Interactions between sugarcane leaf return and fertilizer reduction on soil bacterial network in Southern China red soil" submitted by Liu et al. (2024), on microorganisms aimed to assess the interactions of different  sugarcane leaf return amounts and chemical fertilizer reduction levels on soil bacterial network and relationship between soil properties and bacterial network by using Illumina Miseq high-throughput sequencing technology. The authors found, as expected, that the returning sugarcane leaf to the field increased the diversity of soil bacteria. Interestly, the authors showed that the soil bacterial network were significantly correlated with the soil total K, available N and organic matter contents and EC.

The study is original and compelling, as sugarcane is a crucial crop for various countries. Its cultivation in red soil, which covers approximately 1.13 million km² of southern China—accounting for 11% of the country's land area—makes it even more significant. Furthermore, facilitating research and promoting the application of sugarcane leaf return (SLR) and fertilizer reduction (FR) in the red soil microbial network could yield substantial social, economic, and ecological benefits. Therefore, my recommendation is to approve the study with minor revisions, as follows:

Materials and Methods: Please verify the units used for the soil variables.

The authors mention that the leaves were collected from another experiment. Is there a publication that can be cited?

Instead of including the experiment date, please only mention the period (L112).

Results: Please improve Figures 1, 2, and 3, as they are too small, making it difficult to read the contents. Ensure the captions are self-explanatory.

Figures 05 and 07 are the most problematic; please make the results more visible.

Is Figure 09 really necessary?

Please refine the writing in the discussion section, using more concise and clear English.

For these reasons, my recommendation is to accept the study with minor revisions.

Best regards,

Comments on the Quality of English Language

The quality of the English is moderate; it needs to be written more concisely and directly, mainly on the discussion part.

Reviewer 2 Report

Comments and Suggestions for Authors

Manuscript entitled ‘’ Interactions between sugarcane leaf return and fertilizer reduction on soil bacterial network in Southern China red soil’’ investigate the interactions of different sugarcane leaf return (SLR) amounts and chemical fertilizer levels on soil bacterial network and relationship between soil properties and bacterial network by using Illumina Miseq high-throughput sequencing technology. The authors provide sufficient discussion for the research. The idea of the paper is good, and the research contain interesting results, but the Figures are very small and low resolution. 

There are some grammatical and also space missing in the text and moderate English editing is highly recommended. Please see the comments as below:

Abstract

This section is well written. There is only one point, I would suggest that the authors state the result based on the percentage (only the most important ones), also, keywords should be differ from the title. Make them specific.

Introduction

Line 41: Due to the complexity of soil … -- > before this part, add some previous research on soil microbial network on sugarcane leaf decomposition and highlight the gap of knowledge at them. What was their limitation and …

Paragraph 2: In lines 45-63, since ‘’Uncertainty in calculating the appropriate amount of fertilizer to apply to the field is one of the factors that impact nutrient use efficiency. -- > There is one point: conventional system of farming based on chemical fertilizer, and since the main aim of your study is Optimizing Nutrient and Efficiency, I would suggest that the author add one or two sentences about conventional farming between lines 46-63. (because conventional farming is based on chemical fertilizers). Here is a published work that you can use it https://doi.org/10.3390/su142315870

Line 70: Previous studies ignored --- > add two or three of them here and highlight their limitations and compare them with your research and explain how the current research can fill this gap of knowledge.

Add research hypotheses in the end of the introduction section.

Material and method

Lime 97: soil properties were as follows -- > add soil texture

Results and discussion

Figure 1, 2 and 3: the resolution of the Figures is very low, the legends are invisible, replace them with high resolution ones.

State the changes of core OUTs numbers (Figure 4) with percentage, state the changes in different days based on percentage.

Figure 5: The size of the Figure is very small, the resolution is super low, its not clear and it is not understandable! ATT ALL!!! principal component analysis is a good idea for research like this but the quality of output is very important for illustrating.

Figure 6: like previous ones!!! Both axes are not clear! How should the reader know the treatments of the current Figure and the diversity index?! Provide all the Figures in appropriate size and high resolution.

Again, for Figures 7!

Conclusion

conclusion is well-written with illustrates the lack of knowledge, novelty of work, result of the study. It is better adding suggestions for future studies at the end of conclusion section. 

Comments on the Quality of English Language

Moderate editing of English language required.

Round 2

Reviewer 2 Report

Comments and Suggestions for Authors

No more comments.